# Potential Application of Egg White Peptides for Antioxidant Properties: Perspectives from Batch Stability and Network Pharmacology

**DOI:** 10.3390/foods13193148

**Published:** 2024-10-02

**Authors:** Siwen Lyu, Ting Li, Qi Yang, Jingbo Liu, Ting Zhang, Ting Yu

**Affiliations:** 1Jilin Provincial Key Laboratory of Nutrition and Functional Food and College of Food Science and Engineering, Jilin University, Changchun 130062, China; lyusiwen@163.com (S.L.); jluliting@163.com (T.L.); jluyangqi@163.com (Q.Y.); ljb168@sohu.com (J.L.); tingzhang@jlu.edu.cn (T.Z.); 2Department of Nutrition, The Second Hospital of Jilin University, Changchun 130041, China

**Keywords:** egg white peptides, hydrolysis, antioxidant ability, network pharmacology, oxidative related targets

## Abstract

This study investigated the batch stability of egg white peptides (EWPs) during the enzymatic hydrolysis process, and confirmed the potential application of four crucial four peptides inoxidative damage repair. The results revealed that different batches of EWPs had good stability relating to antioxidant activity. With a similar sequence to confirmed antioxidant peptides, four EWPs (QMDDFE, WDDDPTD, DEPDPL, and FKDEDTQ) were identified withhigh repetition rates, and their potential to repair oxidative damage was investigated. Network pharmacology results showed that these four peptides could regulate the targets related to oxidative damage. Enrichment results demonstrated that these four peptides could influence the targets and pathways related to glutathione transferase activity (enrichment score: 148.0) and glutathione metabolism (*p* value: 9.22 × 10^−10^). This study could provide evidence for the batch stability of hydrolyzed prepared EWPs, and offer theoretical support for the development of antioxidant damage ingredients derived from foods.

## 1. Introduction

Bioactive peptides obtained from egg white were seen as functional ingredients thatcould provide physiological activities such as antioxidant, anti-inflammatory, hypotensive, and hypoglycemic [1]. The way of obtaining egg white peptides (EWPs) and the batch stability of EWP preparation may be one factor affecting product development and application of EWP products. Enzymatic preparation was the most common way to acquire EWPs. However, the selection of enzyme preparations, temperature and pH conditions for hydrolysis could affect the amino acid content, sequence, and functional properties of EWPs. Thus, investigating batch stability during the EWP hydrolysis process could provide vital significance for the EWPs’ acquisition. Recent literature has evaluated the reproducibility of peptides in different batches and identified the sequence by liquid chromatography-mass spectrometry (LC-MS) [2]. However, few papers have been developed on the batch stability of egg white peptides during their preparation. 

In correlation with the inflammatory response, aberrant oxidative stress underlies the pathology of a wide range of chronic conditions [3], such as inflammatory bowel disease (IBD) [4], obesity, and liver diseases. Although clinical drugs are available for the treatment of oxidative damage in humans, the side effects of the drugs limit their widespread application. Thus, people pay attention to the antioxidative ingredients derived from food. A paper has demonstrated the better antioxidative ability of EWPs [5]. The antioxidative ability of EWPs was linked to the peptides’ molecular weight (MW). It was reported that EWPs with an MW lower than 1 kDa showed better antioxidative ability in the egg white hydrolysate [6]. Furthermore, the amino acid constituent and peptides’ sequence could influence the antioxidative ability of EWPs. It was confirmed that many amino acid in the peptides sequences could strengthen the antioxidative ability of EWPs, such as Phe/F, His/H, Cys/C, Pro/P, Tyr/Y, and Met/M [7]. Thus, amino acids and their sequences could indicate the antioxidative ability.

Network pharmacology has been applied as a widely recognized tool and instrument for the analysis of bioactive peptides and their physiological and functional properties. It could discover potential targets and signaling pathways related to a chronic disease [8]. Additionally, network pharmacology analysis is suitable for studying disease intervention mechanisms of functionally active substances with unique components. Currently, there are many studies applying network pharmacology to analyze the potential of bioactive peptide function validation and signaling pathway regulation [9,10,11]. Thus, network pharmacology could apply to the interaction between EWPs and oxidative damage-related targets. To further probe the mechanism of the EWPs’ antioxidative ability, the network pharmacology method was utilized here.

In this study, the batch stability of EWPs was detected by the degree of hydrolysis (DH), amino acid contents, antioxidant activity, and sequence information. After that, four peptides with better reproducibility (QMDDFE, WDDDPTD, DEPDPL, and FKDEDTQ) were identified. Then the potential application of the above four peptides on oxidative damage repair was researched, and the targets and the signaling pathway that the four peptides could regulate were studied. This work could provide a piece of evidence for the batch stability of hydrolyzed prepared EWPs, and offer theoretical support for the development of antioxidant damage of EWPs.

## 2. Materials and Methods

### 2.1. Materials

Red-shelled eggs produced by caged laying hens were purchased at supermarkets and stored in a refrigerator at 4 °C to be used within 3 days. The alcalase used for the EW hydrolysis was bought from Solarbio Biotechnology Co., Ltd (Beijing, China). Sigma Aldrich (St. Louis, MO, USA) provided the Trolox (6-hydroxy-2,5,7,8-tetramethylchroman-2-carboxylic acid), and, ABTS (2,2′-Azinobis-(3-ethyl-benzthiazoline-6-sulphonate)) reagents. The ferric-reducing antioxidant power (FRAP) kit was obtained from Jianchen Bioengineering Institute (Nanjing, China). The AccQ-Tag Ultra Derivatization detection kit was obtained from Waters (Milford, MA, USA).

### 2.2. EWPs Preparation

The EWPs (150 Da < MW <1 kDa) were obtained from fresh EW based on our past method [12]. In brief, the egg white was separated from the eggs using a separator. After that, EW solution (with a concentration of 8%) was obtained. After that, it was heated to denature it, then kept for enzyme hydrolysis (hydrolysis condition: 56 °C and pH 10.0). Alcalase was added to the EW protein at a weight-based ratio of 12%. After three hours, the enzyme was inactivated in the hydrolysate by heating. The supernatant, after centrifugation was separated to obtain fractions with different MW. The 150 Da < MW < 1 kDa fractions were then collected for the following experiments.

### 2.3. Detection of the DH

DH represents the ratio of broken peptide bonds during the enzymolysis process to the inclusive peptide bonds amount in the native protein, which was evaluated based on the pH-stat method, and the formula is as follows [13]:$$\mathrm{DH}\left(\%\right)=\frac{BN_{b}}{Mh_{\mathrm{tot}}\alpha }$$

B and N referred to the dosage of NaOH used during substrate proteolysis and its normal value, respectively; a stood for the average dissociation degree of the a-NH_2_ group in the substrate of protein; M is the mass of the protein in grams; h_tot_ indicated the sum amount of peptide bonds in the substrate of EW protein.

### 2.4. Antioxidant Capacity of the EWPs

#### 2.4.1. ABTS+ Radical Scavenging Capacity Assay

The ABTS experiment of EWPs was carried out in terms of a previous method [14]. The same volume of 2.45 mM potassium persulfate and 7.0 mM ABTS solution were mixed to form an ABTS+ solution, which was then put at room temperature (25 °C) for 12 to 16 h away from light. Prior to use, the above ABTS+ solution was diluted with pH 7.4 phosphate buffer solution (PBS, 75 mM, pH 7.4) to achieve an absorbance at 734 nm of 0.70 ± 0.02. 20 μL. A total of of the appropriate dilution of the sample was then added to 180 μL of the ABTS+ solution in a 96-well plate, the sample was put at room temperature for 10 min, and the absorbance was measured at 734 nm. Absorbance was assayed at 734 nm. Control samples were replaced with 20 μL of distilled water. The ABTS+ scavenging activity of EWPs was determined by the following equation:$$\mathrm{ABTS}\;\mathrm{radical}\;\mathrm{scavenging}\;\mathrm{activity}\;\left(\%\right)=\frac{A_{\mathrm{Control}}-A_{\mathrm{Sample}}}{A_{\mathrm{Control}}}\times 100\%$$

The ABTS radical scavenging capacity of EWPs is displayed as Trolox equivalent (mg TE/100 mg sample) and an IC50 value.

#### 2.4.2. Determination of FRAP

The FRAP capability was determined according to the instructions. The FRAP result was additionally given as Fe^2+^ equivalent (mmol Fe^2+^/100 mg sample).

#### 2.4.3. Determination of Oxygen Radical Absorption Capacity (ORAC)

The ORAC determination is based on previous methods [15]. Preparation of samples and Trolox using PBS. In a 96-well black microplate, a 20 μL sample of Trolox and 120 μL of fluorescein were mixed. Subsequently, using a multimode microplate reader, absorbance was detected at 485 nm and 538 nm for excitation and emission wavelengths, respectively, after mixing with 60 μL of AAPH. The blank control group replaces the sample with 20 μL of PBS. The ORAC values are represented as Trolox equivalents (mg TE/100 mg sample). The calculation formula is as follows:$$\mathrm{ORAC}=\frac{{\mathrm{netAUC}}_{\mathrm{Sample}}}{{\mathrm{netAUC}}_{\mathrm{Trolox}}}\times \frac{M_{\mathrm{Trolox}}}{M_{\mathrm{Trolox}}}$$
$$\mathrm{netAUC}={\mathrm{AUC}}_{\mathrm{Sample}}-{\mathrm{AUC}}_{\mathrm{Blank}}$$

AUC refers to the area under the curve.

A previous method was used here to determine the experimental parameters for LC-MS/MS analysis [12]. The sample was reconfigured with 0.2% formic acid, filled onto a pre-column, and separated on a capillary column by a laser jet. Elution was performed using an Easy nLC 1000 system, and the eluted sample was then puffed into an Orbitrap Elite mass spectrometer. The mass range for the mass spectrometry scan was set at 300–1600 m/z with a resolution of 120,000 and a reaction time of 10 min. For orbital wrap scanning and MS/MS scanning, the auto-gain control target ions were 1e6 and 5e4, respectively. MaxQuant (Version 1.6.1.0) software was used to evaluate the data, and the results were referred to the Gallus source from UniProt (http://www.uniprot.org).

### 2.5. Identification of EWP Sequencse 

A previous method was used here to determine the experimental parameters for LC-MS/MS analysis [12]. The sample was reconfigured with 0.2% formic acid, filled onto a pre-column, and separated on a capillary column by a laser jet. Elution was performed using an Easy nLC 1000 system, and the eluted sample was then puffed into an Orbitrap Elite mass spectrometer. The mass range for the mass spectrometry scan was set at 300–1600 m/z with a resolution of 120,000 and a reaction time of 10 min. For orbital wrap scanning and MS/MS scanning, the auto-gain control target ions were 1 × 10^6^ and 5 × 10^4^, respectively. MaxQuant (Version 1.6.1.0) software was used to evaluate the data, and the results were referred to the Gallus source from UniProt.

### 2.6. Amino Acid Measurement

The determination method of amino acids followed that of our previous research with some modifications [16]. The derivatization of amino acids by 6-aminoquinoline-N-hydroxysuccinimidyl-carbamate (AQC) is relatively uncomplicated and immediate, and the derived amino acids have a long stability time [17]. Amino acid concentrations were analyzed by HPLC. We used 0.1 M HCl to prepare 20 kinds of amino acids into a 0.1 mg/mL solution, then mixed them to obtain an amino acid standard solution. Then, AQC derivatization was carried out manually. First, the EWP powder was dispersed in distilled water (200 mg/mL) and then filtered through a membrane. A total of 10 μL of sample or standard was added to the liquid injection vial. Then, 70 μL of pH 8.2 borate buffer was added into the solution and it was vortexed. Next, 20 μL AQC reagent dissolved in acetonitrile was added, and it was vortexed again. The mixed solution was heated in an oven at 55 ℃ for 10 min and filtered. The Venusil AA column was used from Tianjin Bonna-Angela Technologies Co., Ltd. (Tianjin, China). Solvent A was 10 mM ammonium acetate, and the pH was adjusted to 5.0 by adding glacial acetic acid. Solvent B was 60% acetonitrile by volume. The gradient parameters were set as in our previous study. The flow rate condition was set at 0.6 mL/min and the total run time was 30 min. A 10 μL volume of feed or standard samples was detected using a UV–Vis detector at 248 nm.

### 2.7. Bioinformatic Analysis of EWPs

A multiple sequence alignment study was carried out based on the Clustal Omega tool (version 1.2.4, https://www.ebi.ac.uk/Tools/msa/clustalo/) after redundancies were removed using the Microsoft Excel program to characterize the peptide sequences acquired from the aforementioned analysis bioinformatically [18]. To visually analyze the results obtained, we used use the iTOL (interactive tree of life) online tool (version 5.3, https://itol.embl.de) [19]. The BIO-PEP database predicted the peptides’ functional activity (http://www.uwm.edu.pl/biochemia/).

### 2.8. Network Pharmacology

The structure of the confirmed peptides (QMDDFE, WDDDPTD, DEPDPL, and FKDEDTQ) was created and submitted to the Pharmmapper website (http://www.lilab-ecust.cn/pharmmapper/) to match the oxidative damage related targets [20]. After the targets were collected, the STRING database was utilized to analyze the protein–protein interaction (PPI) network. After that, a topological analysis was constructed based on the Cytoscape software (3.7.1). To further understand the function of targets, enrichment analysis was carried out by the Metascape website (https://metascape.org/gp/index.html), and the figures were depicted by using online and free resource (http://www.bioinformatics.com.cn/).

### 2.9. Statistical Analysis

In the current paper, the statistical analyses were developed by BONC DSS Statistics 25. The differences between two groups and between several groups were evaluated using one-way analysis of variance (ANOVA), accordingly. Tukey’s post hoc test was used to determine significant differences between groups. All experiments in this study were conducted at least three times. Probability values of *p* < 0.05 were considered statistically significant.

## 3. Results and Discussion

### 3.1. DH and Amino Acids Contents of EWPs in Different Batch

Using Alcalase under ideal conditions, based on our previous research, we produced EWPs with a MW of under 1 kDa. The DH of EW was calculated by the pH-stat method. As shown in Figure 1A, the DH of EWPs in six batches was between 29.54% and 36.49%, which indicated a uniform hydrolysis degree of different batches of EWPs. MW is a key parameter reflecting the extent of protein hydrolysis and is further related to the bioactivity of protein hydrolysates. The smaller the MW, the higher the antioxidant capacity of the peptide [21,22]. Thus, we chose a portion of the EWPs (150 Da < MW <1 kDa) to conduct the following experiments. In addition to antioxidant peptides, enzymatic digestion produces other antioxidant-enhancing factors, such as antioxidant amino acids [6,23]. Herein, we detected the free amino acids to further develop the relationship between the antioxidant ability and batch stability of the EWPs. It can be seen from the chromatographic representation of the standards (Figure 1B) that the 20 amino acids were separated accurately and at appropriate levels. Then, we further calculated the amino acid content of the samples based on the peak of the standard amino acids. The content of 20 amino acids in the EWPs of the six batches is shown in Appendix A. The results revealed that the concentrations of 20 kinds of free amino acids in six batches of 200 mg/mL samples were between 0.77 μg/mL and 21.75 μg/mL. To further explore the amino acids contents, we counted the contents of essential amino acids (EAAs) (Figure 1C), non-essential amino acids (NEAAs) (Figure 1D), and total amino acids (TAAs) (Figure 1E) in each batch of EWPs. The data revealed that the EAA content in six batches was between 37.79 ± 1.75 μg/mL and 54.03 ± 2.91 μg/mL, the NEAA content was between 96.74 ± 3.86 μg/mL and 118.60 ± 1.96 μg/mL, and the TAA content was between 138.39 ± 2.24 μg/mL and 162.12 ± 2.20 μg/mL. The above results demonstrated that the different batch has a uniform amino acid content.

### 3.2. Antioxidant Ability Detection of EWPs in Different Batch

As common food-derived functional factors, the antioxidant activity of EWPs has been extensively studied. Redox chemical reaction in vitro is a rapid assay for detecting the antioxidant activity of functional ingredients. According to the different reaction mechanisms, it can be divided into two types: hydrogen atom transfer (HAT) and single electron transfer (SET) reactions. HAT could reflect the capacity of antioxidant peptides to scavenge free radicals by hydrogen supply. SET could reflect the ability of antioxidant peptides to scavenge certain substances by electron transfer. To measure the antioxidant capacity of EWPs objectively and comprehensively, we selected experimental methods of two types of reactions. In the current study, the antioxidant capacity of EWPs was evaluated using the SET reaction mechanism-based FRAP radical scavenging assay and the HAT reaction mechanism-based ABTS+∙ radical scavenging and ORAC assay as indicators. Therefore, three chemical antioxidant activities, including the ABTS radical scavenging rate, FRAP, and ORAC of EWPs, were measured in this study [24].

All six batches showcased ABTS radical scavenging activities with significant dose effects (*p* < 0.05), as shown in Figure 2A. As the concentration of EWPs was 1 mg/mL, the ABTS radical scavenging ability of six batches was between 87.03% and 92.54%. These results showed that EWPs have good antioxidant properties and were stable in the inter-batch. Using the nonlinear regression equation, we calculated the IC50 values (showed in Appendix A) based on the data in Figure 2A. The results revealed that IC50 values of the ABTS+ scavenging ability of six batches were between 0.2611 ± 0.0216 mg/mL and 0.3339 ± 0.0200 mg/mL. In addition, the ABTS+ antioxidant activity of EWPs has been converted to Trolox equivalent values, and the values were from 0.0870 ± 0.0014 to 0.0924 ± 0.0090 mg/mL TE/mg/mL. 

The FRAP assay is based on the reducing ability of a substance to convert ferric tripyridyl triazine (Fe III-TPTZ) to the blue-colored ferrous complex at low pH. Based on the results presented in Figure 2B, the scavenging capacity of FRAP showed a similar trend to ABTS. However, the scavenging ability of FRAP was not as pronounced at the five concentrations measured in the ABTS assay, which may be attributed to the minor changes in sample concentrations. The FRAP values for the six batches were between 0.9380 ± 0.0119 and 1.0207 ± 0.0341 mg/mL Fe/mg/mL, which indicated a uniform trend in different batches of EWPs. 

Finally, we also performed an ORAC assay, a method based on hydrogen atom transfer, which is the most common method used when measuring the egg-derived peptides’ antioxidant capacity [25]. The fluorescence decline curve for the EWPs sample at 0.1 mg/mL is depicted in Figure 3C. The data indicate that the control group had the fastest fluorescence decay rate, and the curve began to stabilize after 70 min, with the relative fluorescence intensity approaching 0. The fluorescence decay rate of the experimental group was slower than that of the control group, indicating that all six batches of EWPs had antioxidant activity. In addition, the fluorescence decay rate of the six groups was relatively consistent and stable at about 130 min. Afterward, the ORAC values of the six groups were between 0.1336 ± 00151 and 0.2255 ± 0.0300 mg/mL TE/mg/mL.

Comparing the three antioxidant assays, the oxygen-radical absorption capacity of EWPs was the best. The above results introduced that EWPs have positive antioxidant activity, as they could donate hydrogen and scavenge free radicals. Chen et al. [26,27] also found the antioxidant properties of EWPs obtained by enzymatic hydrolysis. Our previous research indicated that enzymatic hydrolysis had superior antioxidant activity than untreated EW proteins and could assist in improving exposure to protein active groups [28]. Additionally, the three above experiments indicated that the batch stability of the EWPs had better reproducibility in terms of antioxidant ability. We surmised that the uniform antioxidant ability might be related to the sequence of EWPs in different batches. On this basis, we developed LC-MS/MS and further identified the key peptide sequences that were highly repeatable.

### 3.3. EWPs Sequence Identification by LC-MS/MS

Six batches of EWPs had their peptide sequences identified and examined using LC-MS/MS. MaxQuant (Version 1.6.1.0) software was used to identify 28 peptide sequences for six batches. The length of peptide sequences identified in each group ranged from 6 to 10 amino residues, with m/z ranging from 645.2970 to 1006.5810 Da (Appendix A). The results found that the majority of peptides in the EWPs were derived from ovomucin, ovalbumin, and ovotransferrin. In line with earlier discoveries, sixteen antioxidant peptides from ovalbumin, ovotransferrin, and cystatin were also found [29]. 

Additionally, the results revealed that most of the EWPs were made up of 6-8 amino acid residues, which indicated that our preparation method is relatively adequate. The mass spectrum and structural formula of the EWPs are depicted in Figure 3. These four peptides repeat between batches of EWPs were detected in this study, namely, QMDDFE, WDDDPTD, DEPDPL, and FKDEDTQ. 

The distribution and cleavage sites of the peptide sequences in the native proteins are shown in Figure 4. Lines of various colors represent the lengths of the peptide. The peptide of various batches is represented by various symbols correspondingly. AFKDEDTQ (Ala-Phe-Lys-Asp-Glu-Asp-Thr-Gln) and FKDEDTQ (Phe-Lys-Asp-Glu-Asp-Thr-Gln) were derived from ovalbumin 187-195 and 188-195, respectively. FKDEDTQ was detected in three batches, including 2, 3, and 6. Two ovalbumin-derived peptides, AFKDEDTQ and FKDEDTQ, have similar sequences with the peptide DEDTQAMP. A published paper proved that the peptide DEDTQAMP has a good oxygen-radical absorption capacity, and its ORAC value was 3.14 ± 0.1 μmol TE/μmol [29]. M. Santos-Hernández identified the peptide AFKDEDTQAMP [30], which contains the sequence of two peptides we identified when simulating the in vitro gastrointestinal digestion of the EW protein.

In addition, the peptides DEPDPL derived from ovomucin were identified in our research. The peptide DEPDPL (Asp-Glu-Pro-Asp-Pro-Leu) was derived from ovomucin 687-692, which shares the same sequence as a previously identified antioxidant peptide LDEPDPL. It was identified from ovomucin and proved effective in scavenging free radicals [31]. This result might indicate that the peptide DEPDPL has the most antioxidant properties of the EWPs in our research.

A peptide derived from ovotransferrin, QMDDFE (Gln-Met-Asp-Asp-Phe-Glu), was matched to ovotransferrin 508-517. The peptide QMDDFE was identified in 1, 2, 4, and 6 batches. Ovotransferrin and its derived peptides exhibit high functional qualities as nutraceuticals despite the sequences of the peptides not being mentioned in earlier research. Accumulating evidence suggested that enzymatic hydrolysates of ovotransferrin have greater antioxidant and anticancer activities than the native protein [32]. This condition also suggested that the two peptides we identified may have antioxidant properties.

A peptide’s length, amino acids, and the peptide sequence composition all substantially impact its biological action [33]. The peptide’s maximum length, from our identification results, is 10 amino acids. Free amino acids play a characteristic antioxidant activity according to the properties of their side residues [34]. In addition, negatively charged acidic amino acids (e.g., Asp and Glu) can act as effective electron donors or metal ion chelators, while positively charged basic amino acids (e.g., Arg) can act as influential hydrogen donors [35]. Leu, Pro, Val, and Ala are a few more hydrophobic amino acids useful in enhancing the antioxidant capacity to scavenge free radicals [36]. Another well-known property of antioxidant peptides is the inclusion of aromatic amino acid residues, such as His, Phe, Try, and Tyr [37]. 

Among our identification results, two peptides (DEPDPL and FKDEDTQ) have high repeatability, and the repetition rate was 66.7% and 50%, respectively. The peptide DEPDPL contains one hydrophobic (Leu) residue and three acidic (2 Asp and 1 Glu) residues, accounting for 66.7% of the entire length, which could be effective in radical scavenging activity. The study by [33] found that aromatic amino acids might give protons to electron-deficient radicals to increase free radical scavenging capacity, which provides a basis for this thought. The peptide FKDEDTQ contains one aromatic residue (Phe) and three acidic residues (2 Asp and 1 Glu), accounting for 50% of the entire series length. It has also been shown that the presence of Lys also made contributions to the antioxidant properties. Chen et al. used papain to hydrolyze EW proteins and purify two peptides [38], YLGAK and GGLEPINFN. Both of the two peptides showed stronger DPPH radical scavenging activity than the EW proteins. The amino acid residues Tyr, Gly, Lys, Phe, Glu, and Pro, in peptides were found to increase the antioxidant capacity of peptides. Additionally, with a Lys in the sequence, FKDEDTQ provides evidence that this peptide might have strong antioxidant effects. 

In general, we identified all peptide information from six batches. Based on the length, MW, and amino acid composition of peptides, we speculate that they may have good antioxidant activity, consistent with our antioxidant results. The above results indicate that the EWPs we prepared have specific biological activity and have found potential marker peptides.

### 3.4. Antioxidant Ability Analysis of EWPs in Different Batch

The BIOPEP-UWM database, a valuable tool for detecting the biological activities of peptides, was utilized here to analyze the peptides’ function. The results showed that the EWPs had the following biological activities: ACE inhibitor, antioxidative, and dipeptidyl peptidase IV inhibitor. The antioxidant peptides accounted for 15.79% (Appendix A). A multiple-sequence alignment of more than three biological sequences, such as protein, DNA, or RNA, is known as a multiple-sequence alignment (MSA). Since the sequences in the same column are evolutionarily similar and share an ancestor, the goal is to arrange the identical bases or amino acids from various sequences in that column as much as possible without affecting the sequence order. Non-redundant sequence data sets are essential for bioinformatic analysis. The Clustal Omega tool was then used to examine these peptide sequences through multiple sequence alignment. The respective phylogenetic trees of the EWPs were analyzed based on the iTOL online tool. According to the alignment analysis data, 19 peptide sequences were clustered into three distinct clades, as shown in Appendix A. Without a doubt, the peptide sequences determine their bioactivities. The results showed that, although they have different sequences, they may have similar functional properties. Of the three categories, the first category contains one peptide, the second category includes two peptides, and the remaining sixteen peptides are all classified as category three. Notably, one peptide in category 1 was detected in the 2 and 4 batches. The vast majority of peptides are concentrated in category 3, accounting for 84% of all peptides. Four peptides, FKDEDTQ, DEPDPL, QMDDFE, and WDDDPTD, containing the same part of the sequence as the previous study mentioned, were grouped into category 3, suggesting they have similar properties. Then, we further evaluated the similarity between peptides repeated between batches. The results of multiple sequence comparisons are depicted in Appendix A. The results showed that the four peptides had the same or similar amino acid residues, which might have similar functional activities.

### 3.5. Network Pharmacological Analysis

As a confirmed method used to analyze the bioactive peptides’ functional potential, network pharmacological analysis has been applied to some published papers [39]. In the current study, the four peptides with high reproducibility were selected to analyze the potential targets that could regulate related oxidative damage. The interactions between the EWPs and the targets (Fit score > 6.0) and the PPI network results are shown in Figure 5. Figure 5A revealed that the four peptides WDDDPTD, DEPDPL, FKDEDTQ, and QMDDFE could regulate 17, 13, 10, and 4 targets, which showed the potential to repair oxidative damage. The PPI network and topological analysis are shown in Figure 5B and Appendix A. The results indicated that AKT1, GSTP1, GSTM1, SULT1A1, and GSTA1 could produce more interactions with other targets, which indicated that they might play vital roles in the network. Degree, an important parameter widely recognized, could quantitatively analyze the importance of the target in the network structure [40]. The higher the degree value, the greater the importance of the network. With degrees of 23, 10, 9, 9, and 8, AKT1, GSTP1, GSTM1, SULT1A1, and GSTA1 were seen as the core targets in the PPI. Interestingly, the above targets might link to glutathione metabolism. The PPI results proved the EWPs’ possibility for repairing oxidate damage. However, to evaluate the oxidative damage repair capacity, more analyses were developed subsequently.

To further investigate the targets’ function and the signaling pathway of the EWPs, GO and KEGG analyses were developed, and the results are depicted in Figure 6. The GO analysis was made up of the three following parts: Biological Process (BP), Cellular Component (CC), and Molecular Function (MF) (Figure 6A). The BP results demonstrated that the targets could be affected by EWPs could participate in the process of glutathione transferase activity (enrichment score: 148.0), glutathione peroxidase activity (enrichment score: 100.0), and antioxidant activity (enrichment score: 26.8). In CC part, the results revealed that the targets could affect the cytoplasmic vesicle lumen (enrichment score: 10.2), and secretory granule lumen (enrichment score: 10.2). In the MF results, glutathione derivative metabolic and biosynthetic process were mentioned with a high enrichment score. Additionally, the cellular response to toxic substances’ function is displayed.

The KEGG enrichment results are revealed in Figure 6B. In this part, the top 18 signaling pathways that the targets could influence were listed. As the figure demonstrated, glutathione metabolism was mentioned with a high enrichment score of 57.883 and a *p*-value of 9.22 × 10^−1^. Among the targets regulated by EWPs, six were involved in the glutathione metabolism signaling pathway. Furthermore, a high enrichment score enriched drug metabolism, chemical carcinogenesis, and lipid and atherosclerosis. A published report has proved that the production and metabolism of glutathione might affect the oxidative damage process [41]. Additionally, the glutathione modulation of oxidative stress may be associated with the redox potential, lipid peroxidation, and dysfunctional antioxidant systems. The GO and KEGG analysis results revealed that EWPs had the potential to apply for repairing oxidative damage. On the other hand, network pharmacology was able to preliminarily validate the potential mechanisms by which egg white peptides exert antioxidant activity in vivo, but further cell or animal experiments are needed to demonstrate the actual utility regarding their regulatory targets and signaling pathways.

## 4. Conclusions

Batch stability and functional properties were vital for bioactive peptides’ applications. This paper investigated the potential role of EWPs in oxidative damage repair and integrated the two aspects above. The results showed that the EWPs obtained by Alcalase hydrolysis have promising antioxidant activity and stability between batches. After identification by LC-MS/MS, four peptides repeated between batches were obtained: QMDDFE, WDDDPTD, DEPDPL, and FKDEDTQ. Network pharmacology revealed that AKT1, GSTP1, GSTM1, SULT1A1, and GSTA1 play vital roles in the PPI interaction network. Furthermore, the targets and signaling pathways the four peptides could regulate were analyzed, and the results revealed that the glutathione transferase and peroxidase activity and the glutathione metabolism signaling pathway were enriched with high enrichment scores. The above results indicated that EWPs obtained by Alcalase hydrolysis could maintain the hydrolysis batch stability and obtain four peptides with antioxidant activity. These four egg white peptides may play an antioxidant role by interfering with glutathione metabolism, and they are expected to become active ingredients for repairing oxidative damage. This work provides new perspectives on batch stability and functional properties for the application of EWPs.

## Figures and Tables

**Figure 1 foods-13-03148-f001:**
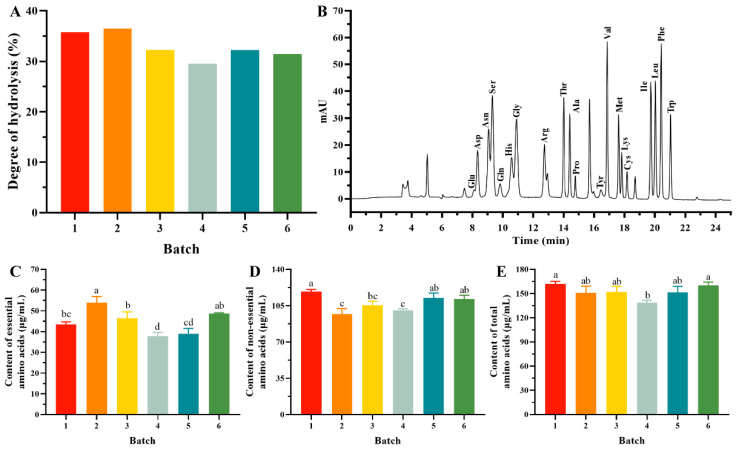
Hydrolysis degree (**A**) of six batches of egg white peptides (EWPs). HPLC chromatograms of a mixed standard solution of 20 amino acids (**B**). Free amino acid contents of EWPs (**C**) and essential amino acids (EAAs). (**D**) Non-essential amino acid (NEAA) contents. (**E**) Total amino acid (TAA) contents. The numbers 1–6 refer to different batches. Different letters mean statistically significant differences at the level of *p* < 0.05.

**Figure 2 foods-13-03148-f002:**
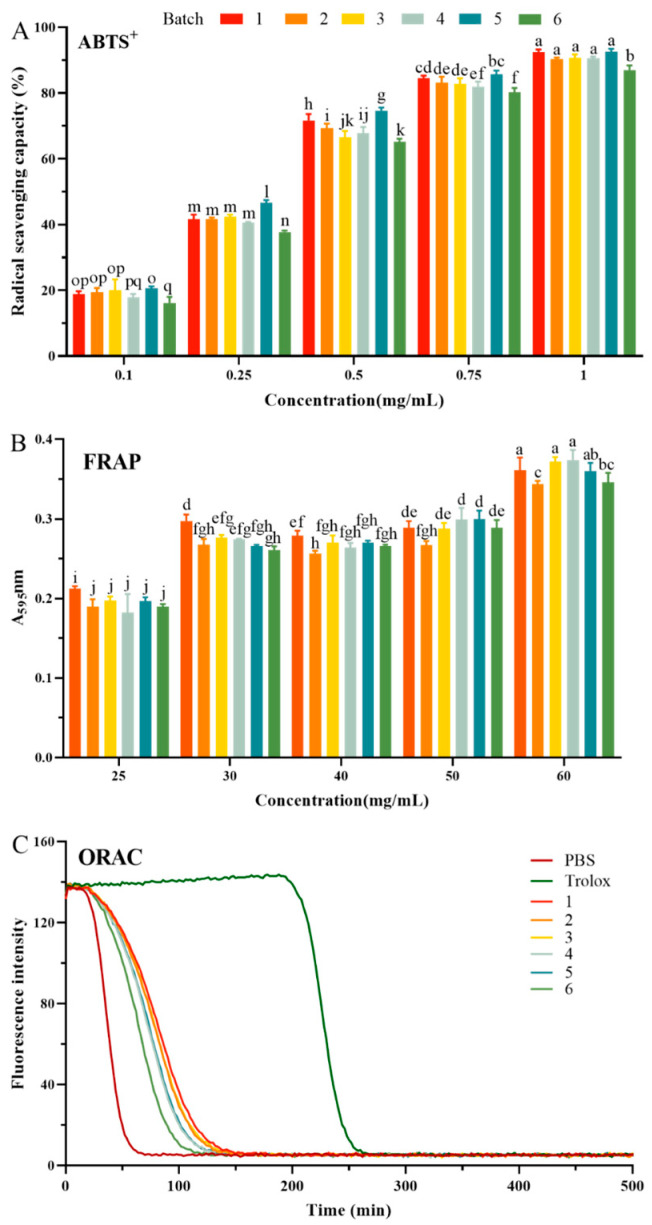
Antioxidant activity in vitro of egg white peptides (EWPs). (**A**) ABTS radical scavenging properties. (**B**) Ferric reducing antioxidant power (**C**) and oxygen-radical absorption capacity. “1–6” means different batches. Different letters mean statistically significant differences at the level of *p* < 0.05.

**Figure 3 foods-13-03148-f003:**
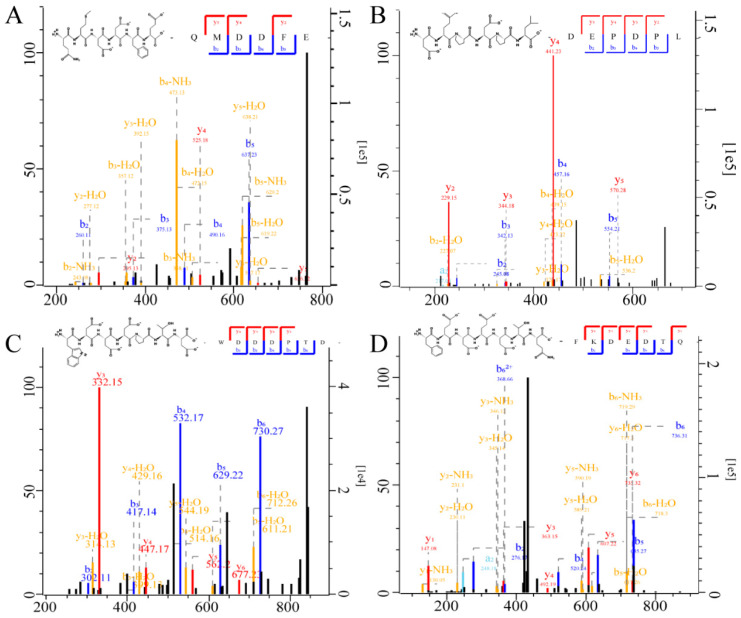
Mass spectrogram and structural formula of peptides (**A**) QMDDFE, (**B**) DEPDPL, (**C**) WDDDPTD, and (**D**) FKDEDTQ with high batch-to-batch repeatability were confirmed according to the LC-MS/MS.

**Figure 4 foods-13-03148-f004:**
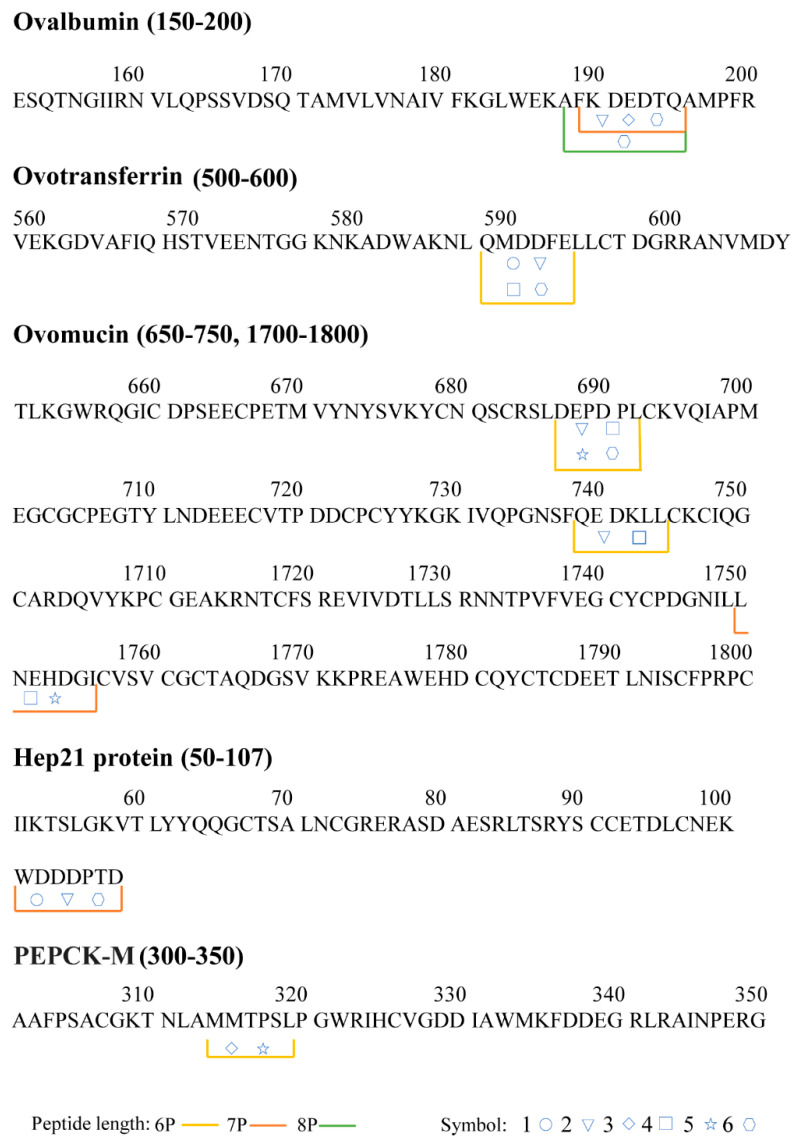
Peptides distribution after LC-MS/MS analysis, “1–6” means different batches. 6P: peptide consisting of six amino acid residues; 7P: peptide consisting of seven amino acid residues; 8P: peptide consisting of eight amino acid residues. The samples of “1–6” were represented by the symbols, respectively.

**Figure 5 foods-13-03148-f005:**
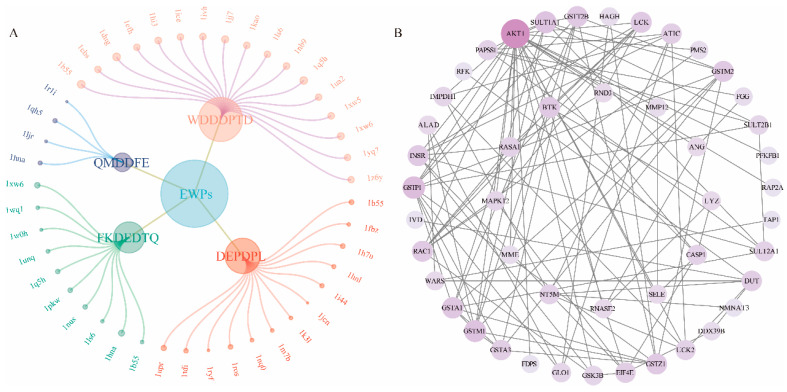
Analysis of network pharmacology results of egg white peptides (EWPs) intervention on oxidative damage-related targets. (**A**) The interaction between EWPs and oxidative damage-related targets. (**B**) The protein-protein interaction (PPI) network of oxidative damage-related targets.

**Figure 6 foods-13-03148-f006:**
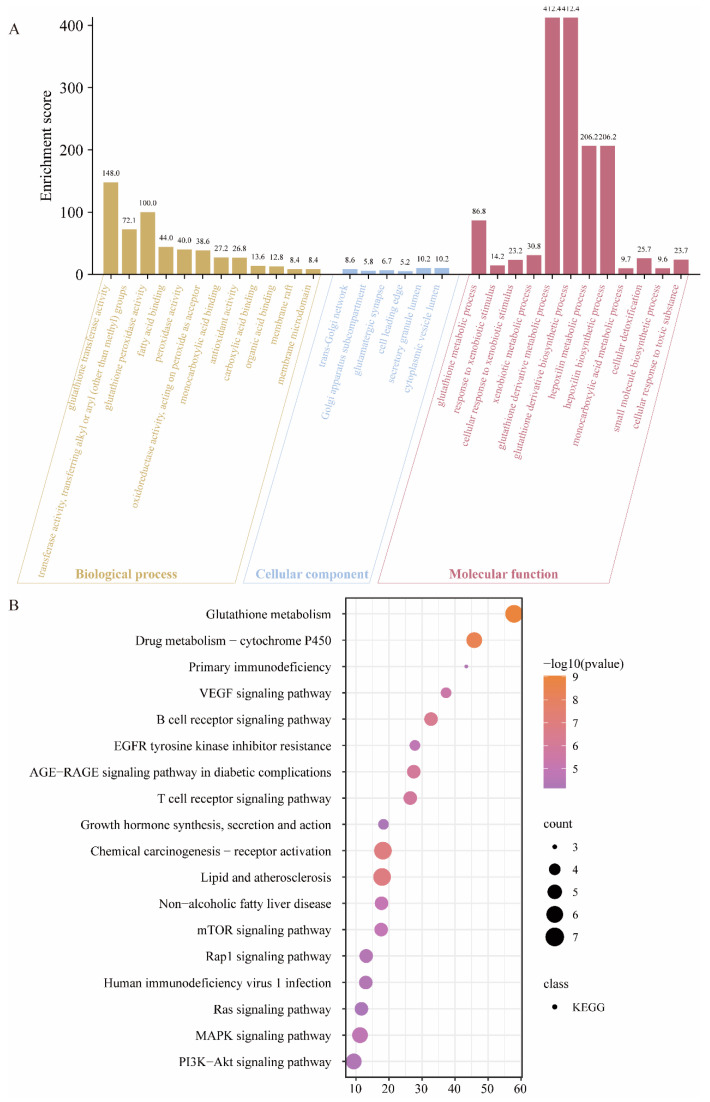
The enrichment analysis results of the oxidative damage-related targets. (**A**) Genetic Ontology (GO) analysis results. (**B**) Kyoto Encyclopedia of Genes and Genomes (KEGG) enrichment analysis results.

## Data Availability

The original contributions presented in the study are included in the article and Appendix A, further inquiries can be directed to the corresponding author.

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
