# Peer review of "Potential Application of Egg White Peptides for Antioxidant Properties: Perspectives from Batch Stability and Network Pharmacology"

_foods, 2024, doi:10.3390/foods13193148_

Round 1

Reviewer 1 Report

Comments and Suggestions for Authors

Comments on the manuscript from Siyen Lyu and co-authors:

1. Title: No evidence regarding the actual 'repair' of oxidative damage has been presented, only evidence from antioxidant properties (based on in vitro testing). Thus, this title should be revised to state 'antioxidant properties' or similar instead of repairing oxidative damage.

2. Introduction: Network Pharmacology is a novel area of research and must be described, with appropriate references from multiple research groups, in its own separate paragraph to inform the readers.

3. Methods: Please provide references for methods, preferably those from different research groups, including those for ABTS, and DH%

4. Eggs: Where did the authors obtain the eggs from? Are these eggs from chickens? If so, please mention the breed if known. Also, the storage and preparation of egg white protein should be described.

5. Statistics: Since ANOVA was used, please state the name of the post-hoc test (e.g., Dunnett's , Tukey's) that was used afterwards to determine the significance, if any, among different groups.

6. The antioxidant and other beneficial properties of the 4 peptide sequences should have been validated by using synthetic peptides with the identical sequence. If not done, a justification must be given.

7. Similarly, the antioxidant a properties should be demonstrated in cell-based assays to show their validity in biological systems. If not done, at least a proper justification should be provided.

Comments on the Quality of English Language

No major concerns

Author Response

  1. Title: No evidence regarding the actual 'repair' of oxidative damage has been presented, only evidence from antioxidant properties (based on in vitro testing). Thus, this title should be revised to state 'antioxidant properties' or similar instead of repairing oxidative damage.

Response: Thank you for your wise opinion. We changed the title into ‘Potential application of egg white peptides for antioxidant properties: perspectives from batch stability and network pharmacology’. For your convenient review, revisions are highlighted in red in the manuscript.

  1. Introduction: Network Pharmacology is a novel area of research and must be described, with appropriate references from multiple research groups, in its own separate paragraph to inform the readers.

Response: Thank you for your advice. We had revised the manuscript based on your opinion. For your convenient review, revisions are highlighted in red in the manuscript.

Revised part (Line 51): Network pharmacology has been applied as a widely recognized tool and instrument for the analysis of bioactive peptides and their physiological and functional properties. It could discover potential targets and signaling pathways related to a chronic disease [8]. Besides, network pharmacology analysis was suitable for studying disease intervention mechanisms of functionally active substances with unique components. Currently, there are more studies applying network pharmacology to analyze the potential of bioactive peptide function validation and signaling pathway regulation [9-11]. Thus, network pharmacology could apply to the interaction between EWPs and oxidative dam-age-related targets. To further probe the mechanism of the EWPs’ antioxidative ability, network pharmacology method was utilized here.

  1. Methods: Please provide references for methods, preferably those from different research groups, including those for ABTS, and DH%

Response: Thank you for the advice. We had added references in the manuscript.

  1. Eggs: Where did the authors obtain the eggs from? Are these eggs from chickens? If so, please mention the breed if known. Also, the storage and preparation of egg white protein should be described.

Response: Thank you for your opinion. Red-skinned eggs produced by caged laying hens were purchased at supermarkets and stored in a refrigerator at 4°C to be used within 3 days (Line 71).

  1. Statistics: Since ANOVA was used, please state the name of the post-hoc test (e.g., Dunnett's , Tukey's) that was used afterwards to determine the significance, if any, among different groups.

Response: Thanks for your fair comments. We had add the analysis methods in the manuscript. Line 174: Tukey's post hoc test was used to determine significant differences between groups.

  1. The antioxidant and other beneficial properties of the 4 peptide sequences should have been validated by using synthetic peptides with the identical sequence. If not done, a justification must be given.

Response: Thank you for your forward-looking evaluation. We are conducting in vitro and in vivo experiments to characterize the antioxidant properties of synthetic egg white peptide sequences, and the relevant results will be submitted soon after processing. The main objective of this thesis research is to initially investigate the antioxidant properties and potential intervention targets of these four peptide sequences, and to provide theoretical support for subsequent data.

  1. Similarly, the antioxidant a properties should be demonstrated in cell-based assays to show their validity in biological systems. If not done, at least a proper justification should be provided.

Response: Thank you for your comment. We are currently carrying out experiments based on hydrogen peroxide-induced oxidative damage in Caco-2 cells, targeting the Nrf2 protein and its related signaling pathways to explore the antioxidant mechanism, and the results of the experiments are being analyzed, and will be submitted after the analysis and finishing.

Reviewer 2 Report

Comments and Suggestions for Authors

In the current study, the authors investigated the AA sequence and antioxidant activity of six batches of egg white peptides; as well as predicted its mechanisms for oxidative damage repair using network pharmacology. Overall, this is an interesting paper with solid results. I only have few minor comments on it:

1. In the material and methods section, the source of egg white should be described. I am a little confused that six batches means the hydrolysis processes was performed independently using the same egg white sample? or the egg white samples were also used from different batches. 

2. In 3.3, the authors identified four peptides consistently exist in the six batches of EWP. However, I would like to know the proportion of these four peptides in the total EWP.

3. Although the Network pharmacological suggested the oxidative damage repair activity of the identified EWP, however, it is unclear the stability of these four peptides in vivo. Therefore, I would doubt the validity of the conclusion. The authors at least should discuss this point in the revised manuscript.

Author Response

  1. In the material and methods section, the source of egg white should be described. I am a little confused that six batches means the hydrolysis processes was performed independently using the same egg white sample? or the egg white samples were also used from different batches. 

Response: Thanks for the opinion. The source of the egg white has been supplemented in the text (Line 71). The same batch of protein solution was applied to the egg white proteins, and the enzymatic process was divided into six groups of separate enzymes.

  1. In 3.3, the authors identified four peptides consistently exist in the six batches of EWP. However, I would like to know the proportion of these four peptides in the total EWP.

Response: Thank you for your advice. We are in the process of conducting quantitative experiments on the four peptides. The purpose of this thesis is to perform qualitative experiments and characterize the antioxidant properties based on mass spectrometry, which will be followed by quantitative experiments and investigation of the antioxidant mechanisms at the cellular level.

  1. Although the Network pharmacological suggested the oxidative damage repair activity of the identified EWP, however, it is unclear the stability of these four peptides in vivo. Therefore, I would doubt the validity of the conclusion. The authors at least should discuss this point in the revised manuscript.

Response: Thank you for your comment. Information about the relationship between preliminary validation of network pharmacology and further experimental investigations has been added in the discussion section (Line 420).

Round 2

Reviewer 1 Report

Comments and Suggestions for Authors

Two comments regarding this revised manuscript:

1. Eggs should be described as 'red shelled', not as 'red skinned'.

2. Future directions including the use of synthetic peptides and testing for antioxidant properties in biologically relevant systems (e.g., cell culture studies) as mentioned in the authors' response to reviewer comments should also be mentioned in the Discussion section.

Comments on the Quality of English Language

Moderate revision is needed.

Author Response

  1. Eggs should be described as 'red shelled', not as 'red skinned'.

Thanks for the opinion. We had changed the word in manuscript (Line 71).

  1. Future directions including the use of synthetic peptides and testing for antioxidant properties in biologically relevant systems (e.g., cell culture studies) as mentioned in the authors' response to reviewer comments should also be mentioned in the Discussion section.

Thank you for the advice. The discussion was added into the manuscript (Line 420-423).

Reviewer 2 Report

Comments and Suggestions for Authors

The author has responded to most of my questions, but I still believe that the key issue regarding the proportion of peptides has not been adequately addressed.

For instance, if the identified peptide only accounts for 0.001% of the total peptides samples, how it could be the key molecules to explain the overall antioxidant activity. And thus it could not support the Network pharmacological  analysis. 

Since the author is currently conducting the related experiments, I would like to recommend acceptance after the authors complete the experiments and include the relevant results.

Author Response

The author has responded to most of my questions, but I still believe that the key issue regarding the proportion of peptides has not been adequately addressed.

For instance, if the identified peptide only accounts for 0.001% of the total peptides samples, how it could be the key molecules to explain the overall antioxidant activity. And thus it could not support the Network pharmacological analysis.

Since the author is currently conducting the related experiments, I would like to recommend acceptance after the authors complete the experiments and include the relevant results.

Answer: Thank you for your pertinent comments. The content of the peptide sequence is indeed a very worthwhile consideration, but it is not the focus of this thesis. The focus of this study is the inter-batch characterization of egg white peptide sequences and the application of in vitro chemistry and network pharmacology for the resolution of antioxidant mechanisms. It is in future studies that we will quantitatively analyze the peptide sequences and biologically validate the antioxidant properties using cellular or animal experiments. These two articles have different approaches and focuses.

Round 3

Reviewer 1 Report

Comments and Suggestions for Authors

Thank you for accepting the suggestions.

Comments on the Quality of English Language

It is acceptable.

Reviewer 2 Report

Comments and Suggestions for Authors

The authors have reiterated the reason why they do not quantify the identified peptide. I can barely accept their explanation but would like to endorse its acceptance since they already addressed all other questions.